# Tabletop 360-Degree Three-Dimensional Light-Field Display Based on Viewpoint-Fitting Encoding Algorithm for Reducing Facet Braiding

**DOI:** 10.3390/mi14010178

**Published:** 2023-01-10

**Authors:** Peiren Wang, Jinqiang Bi, Zilong Li, Binbin Yan, Zhengyang Li, Xiaozheng Wang, Li Liu

**Affiliations:** 1Tianjin Research Institute for Water Transport Engineering, Ministry of Transport, Tianjin 300456, China; 2State Key Laboratory of Information Photonics and Optical Communications, Beijing University of Posts and Telecommunications, Beijing 100876, China; 3Research Department of Wireless Network, Huawei Technologies, Shanghai 201206, China; 4Chipone Technology (Beijing) Co., Ltd., System Engineering Department, Beijing 100176, China

**Keywords:** facet braiding, depth of field, viewpoint-fitting encoding algorithm, tabletop 3D light-field display

## Abstract

Since the effect of the facet braiding phenomenon, the display quality of reconstructed image degrades with increasing depth of field in tabletop three-dimensional light-field display. Here, to analysis the facet braiding, the imaging process of the tabletop 360-degree three-dimensional light-field display based on conical lens array is mathematically modeled. A viewpoint-fitting encoding algorithm is proposed to reduce the effect of the facet-braiding phenomenon and improve the range of depth of field, which is optimized to form the best synthetic encoded image by fitting the reconstructed image seen by the simulated human eye to the parallax image captured at the corresponding location. The effectiveness of the proposed optimization algorithm is verified by simulation analysis and optical experiments, respectively. In the experiment, the clear depth of field range of the display system is increased from 13 cm to 15 cm, and the visualization effect of the reconstructed three-dimensional image is enhanced.

## 1. Introduction

As the great innovation of three-dimensional (3D) display technology and the urgent needs of information display, the research of tabletop 3D display has attracted more attention and made great progress [1,2,3]. The conventional tabletop 3D displays, such as holographic 3D display [4,5,6], integral imaging 3D display [7,8,9], volumetric 3D display [10,11,12], light-field 3D display [13,14,15] and Microstructure-based 3D display [16,17,18], can provide more information and more realistic 3D scenes. However, the inherent drawbacks limit the application of tabletop 3D displays, for example, the shallow depth of field, the low viewing resolution and the viewing jump. With the rapid development of the computer graphics and computer vision, massive 3D date at high resolution can be acquired, which is widely used in electronic map [19,20]. Unfortunately, most of the existing tabletop 3D display systems cannot meet the expectation of high-quality terrain rending technology. Nevertheless, the tabletop 3D light-field display provides a perfect solution for electronic sand table, which can present all depth cues on the table and light rays distributed with variable brightness and different colors in different directions [1]. In our previous work, a tabletop 360-degree 3D light-field display based on aspheric conical lens array has demonstrated, which provide a ring-shaped viewing range to improve viewpoints utilization [21]. Therefore, the observer can access multiple pixels through a lens to improve viewing resolution. However, the image quality deteriorates as the depth of field increases. The quality deterioration is mainly due to the facet-braiding phenomenon caused by the reconstruction mechanism of the synthetic coded image based on the back-ward ray tracing algorithm.

Insufficient depth of field seriously limits the development of the tabletop 3D light-field display [22], and the facet-braiding phenomenon is the major factor leading to this problem [23], which appears due to the conflict between the optical focusing ability of lens array and the characteristics of rays intersecting for 3D scene reconstruction. The facet-braiding phenomenon is common in integrated imaging display systems [24], for instance, the imaging process based on point object is shown in Figure 1, where L is the distance from the reference image plane to the lens array. In the pickup process, the intensity and the directional information of dots located at different spatial positions are recorded on CCD. In the display process, the reconstructed spatial dots can only be matched correctly at the record position where they originally were. The pixels on the LCD are imaged on the reference image plane. Thus, the yellow dot can be reconstructed as a complete dot on the reference image plane, and the green dot will be reconstructed a speckle instead of a dot. Due to the natural adjustment of the visual system, the human eye will automatically focus on the reference image plane [25]. As a result, a bright yellow dot and a slightly dark green dot will be seen. Hence, the facet-braiding phenomenon appears on the reference image plane with the 3D scene, which leads to the reconstruction of a fuzzy spot instead of a legible dot. Recently, the facet-braiding phenomenon is analyzed [26,27] and the influences on depth of field is discussed in some works [28]. However, there is no specific optimization algorithm proposed to address the degradation problem of 3D display image quality caused by the facet braiding phenomenon.

In this paper, using the tabletop 360-degree 3D light-field display system based on the conical lens array as the hardware, the effect of the facet-braiding phenomenon lending to depth of field limitation is dissected. By mathematically modeling the imaging process of the 360-degree tabletop 3D light-field display based on conical lens array, a viewpoint-fitting encoding algorithm is proposed to significantly reduce the facet-braiding phenomenon and improve the depth of field, which is optimized to form the best synthetic encoded image by fitting the reconstructed image seen by the simulated human eye to the parallax image captured at the corresponding location. In the experiment, a high-quality tabletop 360-degree 3D light-field display system with high resolution and large depth of field is realized. The clear depth of field range of the display system is increased from 13cm to 15cm by means of the proposed optimization algorithm, and the visualization effect of the reconstructed 3D image is enhanced.

## 2. Principle

### 2.1. Analysis of the Facet Braiding Phenomenon

To understand the facet-braiding phenomenon more intuitively and naturally, the previous 360-degree tabletop 3D light-field display system is drawn, which is composed of the holographic functional screen (HFS), the conical lens array, LCD panel and the collimating backlight array [29], as shown in Figure 2. The collimating backlight array can make each elemental image on the LCD panel in the form of directional beams instead of scattered beams, which ensure low crosstalk between viewpoints. The pixels in each elemental image can be arranged in a shape of circular ring on the HFS by the special conical lens array. In addition, the conical lens array combined with the HFS can not only reduce the aberration and distortion of the displayed image, but also eliminate the gap between the lenses. However, as the display depth of the reconstructed image increases, the facet-braiding phenomenon leads to the deterioration of the display image quality, which cannot provide a satisfactory visual effect. Therefore, the farther the reconstructed plane is from HFS, the more serious the facet-braiding phenomenon is.

Actually, the viewed 3D image is spliced by a serious of sub-images observed through each a lens. The reference plane is the conjugate plane of the lens array. The pixels on the liquid crystal panel will be imaged on the reference plane, and the reconstruction plane is the actual depth plane of the captured 3D object. If the object is far away from the reference plane in the pickup stage, the sub-images viewed by human eye will not be spliced correctly in the display stage. This is called facet-braiding phenomenon [24,25], which results in the limited depth of field and deterioration of display quality. When the reference plane is inconsistent with the reconstructed plane, the facet-braiding phenomenon appears. As the distance between the two planes increases, the facet-braiding phenomenon becomes more obvious. According to the positional relationship between the two planes, the facet-braiding phenomenon is divided into two types. When the reconstructed plane is far away from the reference plane, the facet braiding is called direct displacement, as shown in Figure 2a. When the reconstructed plane is located between the reference plane and the lens array, the facet braiding is called inverse displacement, as shown in Figure 2b. The facet-braiding phenomenon is the decisive factor that limits the depth of field of 3D display system. Although many works have been conducted on it in recent years, there are few specific solutions for the phenomenon to improve the depth of field and display quality.

### 2.2. Efficient Solution to the Facet Braiding Phenomenon

As shown in Figure 3a, similar to the most 3D light-field display systems, the synthetic image is access by the parallax images captured by the camera array in the pickup process and the pixels-mapping algorithm based on the backward ray-tracing technology, which can be loaded into the display system to reconstruct the recorded 3D scene.

To reduce the deterioration of displayed image caused by the facet braiding phenomenon, the 360-degree tabletop light-field display system based on the conical lens and HFS is mathematically modeled. According to the modeled display system, each viewpoint perspective based on different positions of the observer in the space can be accurately obtained. In terms of system structure and the arrangement of viewpoints, the viewing area is divided into *M*^2^ sub-areas, and each sub-area also corresponds to the position of the camera in the pickup process. According to Figure 3b, the viewpoint perspective $g_{\left(i,j\right)}$ seen by the observer at different positions can be modelled as:(1)$$g_{\left(i,j\right)}=D_{\left(i,j\right)}WS\;(i\in [1,2\ldots ,M],j\in [1,2\ldots ,M])$$
where *M* is the number of pixels in the elemental image along the horizontal or vertical direction. *S* (size of [*M^2^*U*V* × 1]) denotes the synthetic coded image loaded on the LCD panel. The matrix *W* (size of [γ*^2^M^2^*U*V* × γ*^2^M^2^*U*V*]) represents the light modulation of the lens array, which is the parallax shift of the lens to solve the depth reversed. The matrix $D_{\left(i,j\right)}$ represents how the light information on the HFS is subsampled by the viewer, and the sampling information is adjusted with a different viewing position. $g_{\left(i,j\right)}$ represents the viewpoint perspective observed by the human eye at different positions within the viewing area, which is spliced by U × *V* different facet images from the U × *V* lens array. The facet image is composed of γ*M* × γ*M* pixels covered under a lens. γ is the sampling factor, which ranges from 0 to 1/2. According to the geometric relationship in Figure 3b, the sampling factor γ can be obtained by the Equation (2).
(2)$$\gamma =\frac{PH}{L(H+L)(tan\varphi_{outer}-\tan\varphi_{inner})}$$
where *P* is the diameter of the conical lens, H represents the distance between the viewing plane and the HFS, and *L* represents the distance between the HFS and the conical lens array. The inner ring with $\varphi_{inner}$ = 20° is constructed by the central pixels of the elemental image and the outer ring with $\varphi_{\mathrm{outer}}$ = 35° is constructed by the edged pixels of the elemental image.

According to the mathematic analysis and the synthetic coded image, the viewpoint perspective $g_{\left(i,j\right)}$ observed from different positions can be calculated by the Equation (1) to simulate the imaging process of the light-field display system. As shown in Figure 4, the traditional lens imaging method enlarges the pixels of the elemental image, and then superimposes them on the HFS. The image blur caused by the superposition process of adjacent pixels is the facet braiding phenomenon. Therefore, the optimal synthetic image *S* can be fitting by combining the reconstructed image $g_{\left(i,j\right)}$ observed by the human eye with the parallax image $G_{\left(i,j\right)}$ captured by the camera at the corresponding viewing position during the pickup stage. Converting it into the mathematical relation expression, the maximum likelihood estimation of the optimized synthetic image *S* can be calculated in Equation (3).
(3)$$S=\underset{S}{\arg\min}\left[\sum_{i=0}^{M}\sum_{j=0}^{M}\rho (g_{\left(i,j\right)},G_{\left(i,j\right)})\right]$$
where *ρ* represents the “distance” between the reconstructed image $g_{\left(i,j\right)}$ observed by the simulated human eye and the parallax image $G_{\left(i,j\right)}$ captured by the virtual camera. To obtain the estimation results of the synthetic image, *S* is set to *L*_2_ norm. Therefore, the following equation can be obtained by bringing Equations (1)–(3).
(4)$$\underset{S}{\arg\min}\left[\sum_{i=0}^{M}\sum_{j=0}^{M}{\|D_{\left(i,j\right)}WS-G_{\left(i,j\right)}\|}_{2}^{2}\right]$$

Due to the solution of the above indeterminate equation is uncertain, the optimized synthetic image *S* is solved using the gradient descent method, and thus the high-quality reconstructed 3D image with minimal facet braiding is realized. In addition, the coded image [21] is adopted as the initial value of the synthetic image *S* to improve the convergence efficiency of the proposed algorithm. The core idea of the proposed algorithm can precisely match the correspondence between pixels on the display panel and viewpoint information without additional hardware cost, weakening the deviation effect of pixels on the reconstruction plane farther away from the imaging plane and improving the imaging quality within the reconstruction plane of depth of field, thus expanding the range of the ultimate depth plane viewed by the human eye.

## 3. Simulation Analysis and Contrast Experiment

Relying on the previously demonstrated tabletop 360-degree 3D light-field display system, the viewpoint-fitting coding optimization algorithm is combined to expand the rang of depth of field and improve the display quality. To further evaluate the quality of the reconstructed 3D images, the following simulations were performed to compare the results before and after the algorithm optimization. It is well-known that the reconstructed display image observed by human eye is formed by stitching a series of sub-images in each lens unit. The effectiveness of the proposed algorithm was evaluated by introducing the structural similarity index measure (SSIM) to calculate the value of the viewpoint perspectives with different angles. As shown in Figure 5, the SSIM value of the reconstructed image with the proposed optimized algorithm is obviously higher than that without the optimized algorithm, which illustrates that the optimized reconstructed image has high similarity.

Figure 6 shows the actual imaging results of the traditional reconstruction method compared with the proposed optimization algorithm. As shown in Figure 6a, the reconstructed 3D image appears blurred at the large depth-of-field plane due to the facet braiding phenomenon, which limits the range of clear display image. However, by introducing the viewpoint-fitting encoding algorithm, the clarity of the reconstructed display image is significantly improved at the large depth of field plane. The clear depth of field range of the display system is increased from 13 cm [21] to 15 cm.

## 4. Experimental Results

To verify the effectiveness of proposed method, relevant experiments are carried out. In the tabletop 360-degree 3D light-field display system, the size of the LCD panel used to load the synthetic encoded image is 27 inch with a resolution of 3840 × 2160. The number of LEDs, Fresnel lenses, elemental images and conical lenses are the same. The fixed structure of the aluminum plate using a computer numerical control machine to uniformly punch 46 × 26 holes with a diameter of 10 mm and a distance of 13 mm between the centers of adjacent holes, and put the conical lens array into the holes in turn. A HFS with a 5° diffusion angle is used to eliminate the gap between adjacent conical lenses. The detailed parameters of the system are shown in Table 1.

With the advantages of annular viewing of 3D scenes and intuitive 3D visual perception, the electronic map is one of the most ideal directions for tabletop 3D light-field display. The synthetic encoded image loaded on the LCD panel is optimized by the proposed viewpoint-fitting encoding algorithm, different perspectives along the annular direction of the 3D electronic map are shown in Figure 7 and Appendix A. The display results illustrate that the proposed optimization method can significantly reduce the facet-braiding phenomenon and provide a clear 3D display with correct occlusion relationship and high quality for observers in different viewing positions.

Another prominent application of the tabletop 3D light-field display is for medical diagnosis and biomedical education. Figure 8 shows the reconstructed 3D display images of the medical data for human organ with different angles. The reconstructed 3D human organ of heart and sternum with correct geometric relationships and accurate depth clues can be observed. Thus, the tabletop 3D light-field system with proposed optimization algorithm can provide natural and clear 3D scenes, intuitively.

## 5. Conclusions

In a tabletop full-parallax 3D light-field display based on lens array, the facet-braiding phenomenon leads to blurred images in a large depth of field plane, which is a major factor limiting the depth of field. In this paper, the imaging process of the tabletop 360-degree three-dimensional light-field display based on conical lens array was mathematically modeled to analyse the facet braiding phenomenon. A viewpoint-fitting encoding algorithm was proposed to reduce the effect of the facet-braiding phenomenon and improve the range of depth of field. The effectiveness of the proposed optimization algorithm was verified by simulation analysis and optical experiments, respectively. With our proposed algorithm, a tabletop 360-degree 3D light-field display system with large depth of field was experimentally demonstrated to present the reconstructed high-quality 3D images of the military and medicine. The clear depth of field range of the display system increased from 13 cm to 15 cm, and the visualization effect of the reconstructed 3D image was enhanced.

## Figures and Tables

**Figure 1 micromachines-14-00178-f001:**
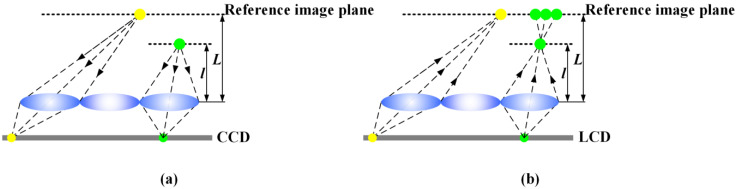
Facet braiding effect for point objects in (**a**) pickup process and (**b**) display process.

**Figure 2 micromachines-14-00178-f002:**
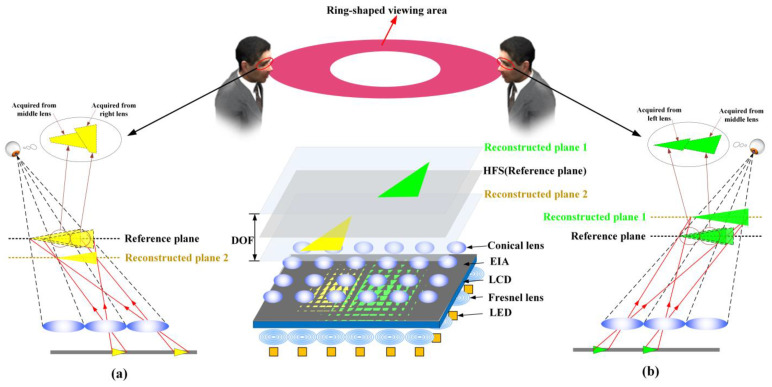
Facet-braiding phenomenon: (**a**) Inverse displacement, (**b**) direct displacement.

**Figure 3 micromachines-14-00178-f003:**
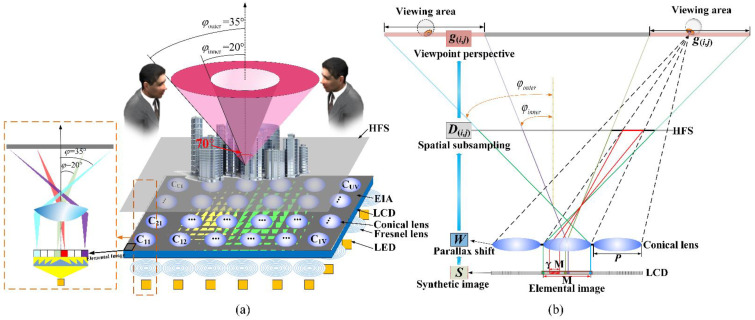
(**a**) The configuration of 360-degree 3D light-field display system. (**b**) Modeling the 3D light-field display.

**Figure 4 micromachines-14-00178-f004:**
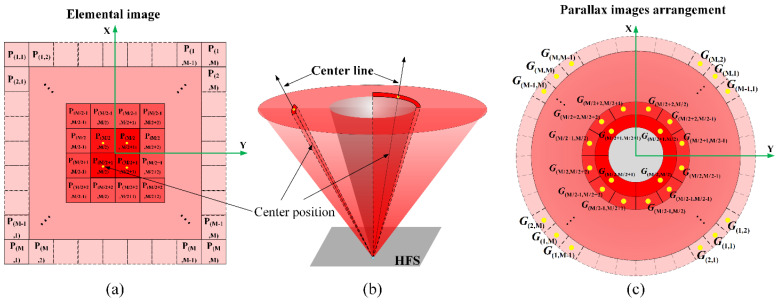
(**a**) Pixels of elemental image displayed on the LCD. (**b**) Center lines of the light rays emitted from HFS. (**c**) The distribution of the parallax images array captured by the virtual camera array.

**Figure 5 micromachines-14-00178-f005:**
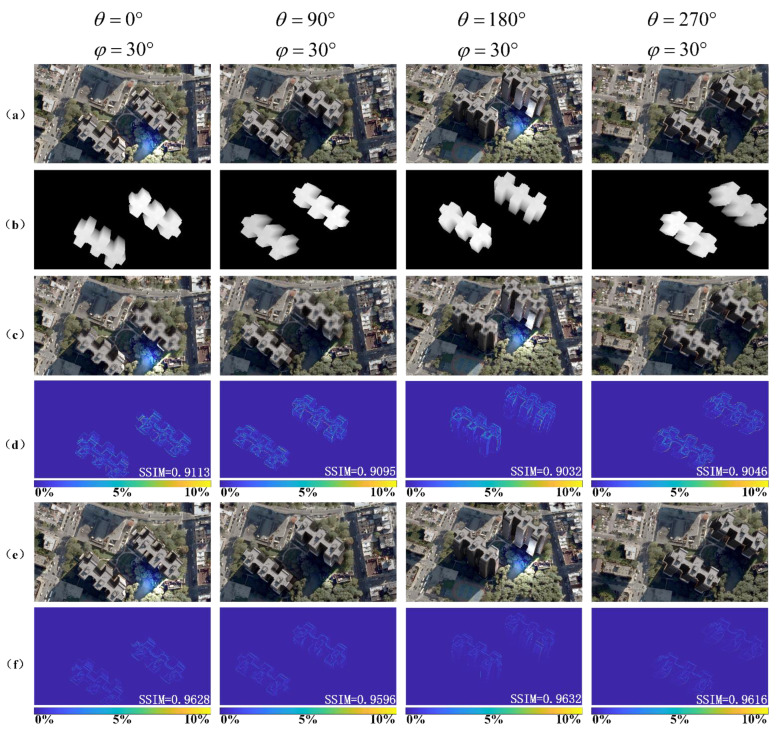
Simulation results: (**a**) Perspectives captured from different angles; (**b**) depth maps of different perspectives; (**c**) simulation results of the reconstructed image without optimized algorithm; (**d**) the SSIM values without optimized algorithm; (**e**) simulation results of the reconstructed image with proposed optimized algorithm; (**f**) the corresponding SSIM values with proposed optimized algorithm.

**Figure 6 micromachines-14-00178-f006:**
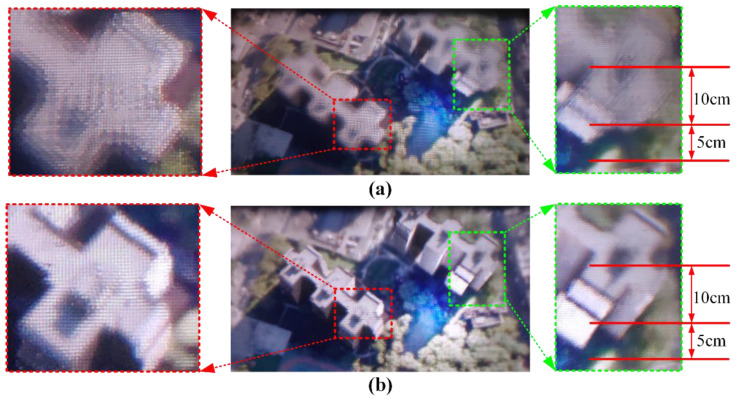
The comparison of reconstructed 3D images. (**a**) Before optimized algorithm; (**b**) after proposed optimized algorithm.

**Figure 7 micromachines-14-00178-f007:**
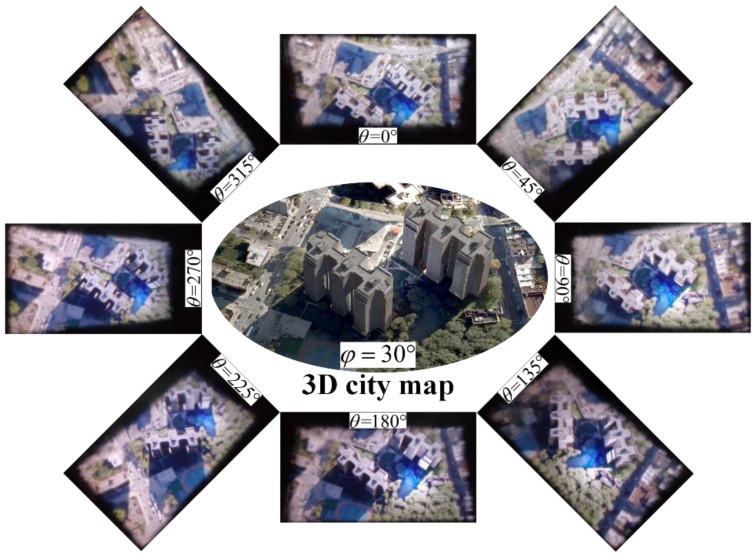
Different perspectives along the annular direction of the 3D electronic map (Appendix A).

**Figure 8 micromachines-14-00178-f008:**
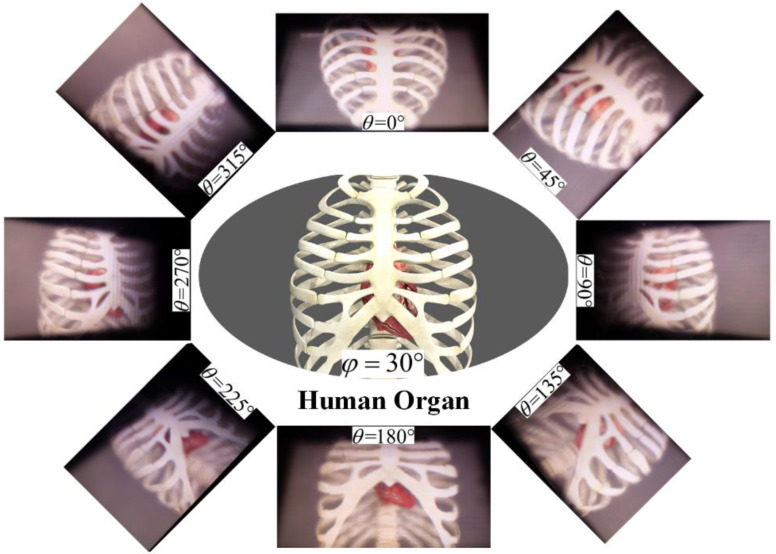
Different perspectives along the annular direction of the 3D human organ of heart and sternum (Appendix A).

**Table 1 micromachines-14-00178-t001:** The specific parameters of tabletop 360-degree 3D light-field display system.

Parameter		Detail
Backlight	The number of LEDs	46 × 26
	The diameter of LED	2 mm
	The number of Fresnel lenses	46 × 26
	The diameter of Fresnel lenses	10 mm
	The focus of Fresnel lenses	15 mm
Display panel	The size of LCD panel	27 inch
	The resolution of LCD panel	3840 × 2160
Lens array	The number of conical lenses	46 × 26
	The diameter of conical lenses	10 mm
	The pitch of adjacent conical lenses	13 mm
HFS	The horizontal divergence angle of HFS	5°
	The vertical divergence angle of HFS	5°
Annular viewing angle	The inner ring angle	20°
	The outer ring angle	35°
The distance between the HFS and the conical lens array		200 mm

## Data Availability

Not applicable.

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
