# Peer review of "Tabletop 360-Degree Three-Dimensional Light-Field Display Based on Viewpoint-Fitting Encoding Algorithm for Reducing Facet Braiding"

_micromachines, 2023, doi:10.3390/mi14010178_

Round 1

Reviewer 1 Report

Refer to the attached file

Reviewer 2 Report

The authors propose a tabletop 360-degree three-dimensional light field display based on viewpoint fitting encoding algorithm for reducing facet braiding. The effect of facet braiding phenomenon is analyzed in detail for the depth of 3D images. The purpose of holographic functional screen for the tabletop three-dimensional display system is to eliminate the gaps between 3D pixel and make the reconstructed 3D image smooth and natural. The holographic function screen affects the depth of field of the 3D image. The authors propose a viewpoint fitting algorithm that is believed to eliminate the facet braiding phenomenon. I do not think there is a necessary relationship between the viewpoint fitting algorithm and the facet braiding phenomenon. The viewpoint fitting algorithm may reduce the facet braiding phenomenon, but it does not eliminate the facet braiding phenomenon. Therefore, I think the novelty of this manuscript does not meet the requirements for publication.

1. This manuscript only makes minor adjustment to the algorithm compared to “X. Yu, X. Sang, X. Gao, B. Yan, D. Chen, B. Liu, L. Liu, C. Gao, and P. Wang, 360-degree tabletop 3D light-field display with ring-shaped viewing range based on aspheric conical lens array, Opt. Express, 27, 26738-26748,2019”. The method of arranging the pixels in the elemental image array, as well as the 2D display screen, the lens array and the holographic functional screen used in the experiments, remains unchanged.

2. The measurement method of 3D image depth is not clearly described, and the results in Figure 6 are not convincing.

Reviewer 3 Report

The authors propose a viewpoint fitting encoding algorithm for the tabletop 360-degree 3D light field display to increase the depth of field. The experimental results give the simulation 3D images before and after optimization. The core of the proposed encoding algorithm is Eqs. (3) and (4). However, the direct relationship between the two equations and the reduction of the face braiding is not clearly analyzed. Following are some comments.

1. The core idea of this manuscript is confusing. The facet braiding represents the inconsistency between the lens imaging plane and the 3D point fusion plane, and it is inevitable. The facet braiding determines the depth of field of the integral imaging 3D display. It seems that the purpose of the proposed algorithm is extending the depth of field, rather than reducing the facet braiding. The authors also do not give a detailed analysis of how the problem of face braiding is solved by using the viewpoint fitting encoding in Lines 151-169.

2. It seems that the proposed algorithm is only applicable to the simulated integral imaging display of viewing angle priority, not the optical integral imaging display. The core of the optimization is making the reconstructed image approximate the ideal clear parallax image only on the central depth plane in simulation. And the problem of the face braiding is analyzed for the image on the central depth plane. In optical integral imaging display, the human eye will not automatically focus only on the central depth plane. The authors should state this limitation.

3. Please provide a comparison result of the synthetic image S before and after optimization.

4. Eqs. (3) and (4), “P(i, j)” should be “G(i, j)”, and “T” should be “W”, right?

5. Fig. 3(b), “Element image” should be “Elemental image”. Please correct it.

6. Line 153, “light filed” is incorrect.

7. The figure caption of Fig. 4(c) is confusing. What does parallax images array mean?

Round 2

Reviewer 2 Report

I am very sorry that I still stand by my previous review comments. I think the novelty of this manuscript does not meet the requirements for publication. I read the author's response in detail and my concerns are not answered to satisfaction.

1.      The manuscript does not explain how the viewpoint fitting algorithm solves the facet braiding phenomenon. I think even though the viewpoint fitting algorithm can accurately match pixels on the display panel and the viewpoint information, the facet braking phenomenon still exists.

2.      The manuscript does not explain how the viewpoint fitting algorithm enhances the depth of field of 3D images. I don't find any corresponding theory to support it. The author's reply also does not explain.

3.      In addition, the measurement method of 3D image depth is not clearly described, and the results in Figure 6 are not convincing. The author's reply also does not explain, and the authors does not replace the new measurement method.

Reviewer 3 Report

The authors have addressed my previous comments. Therefore, I believe this manuscript can be published in Micromachines.